# Comparison of Subjective and Objective Assessments on Improvement in Gait Function after Carotid Endarterectomy

**DOI:** 10.3390/s20226590

**Published:** 2020-11-18

**Authors:** Tatsuhiko Takahashi, Shunrou Fujiwara, Suguru Igarashi, Toshihiko Ando, Kohei Chida, Masakazu Kobayashi, Kenji Yoshida, Takahiro Koji, Yoshitaka Kubo, Kuniaki Ogasawara

**Affiliations:** Department of Neurosurgery, Iwate Medical University, 1-1-1 Idaidori, Yahaba 028-3695, Iwate, Japan; dyspro7@gmail.com (T.T.); shunfuji@iwate-med.ac.jp (S.F.); guru-guru-guru@hotmail.co.jp (S.I.); toshihiko.a.12@gmail.com (T.A.); kchida@iwate-med.ac.jp (K.C.); kobamasa@iwate-med.ac.jp (M.K.); kenyoshi@iwate-med.ac.jp (K.Y.); thkoji@hotmail.com (T.K.); yokubo@iwate-med.ac.jp (Y.K.)

**Keywords:** carotid endarterectomy, gait, tri-axial accelerometer, stride time, cadence, ground floor reaction

## Abstract

The purpose of the present study was to determine whether objective gait test scores obtained using a tri-axial accelerometer can detect subjective improvement in gait as determined by the patient after carotid endarterectomy (CEA). Each patient undergoing CEA for ipsilateral internal carotid artery stenosis determined whether their gait was subjectively improved at six months after CEA when compared with preoperatively. Gait testing using a tri-axial accelerometer was also performed preoperatively and six months postoperatively. Twelve (15%) of 79 patients reported subjectively improved gait. Areas under the receiver operating characteristic curve for differences between pre- and postoperative test values in stride time, cadence, and ground floor reaction for detecting subjectively improved gait were 0.995 (95% confidence interval (CI), 0.945–1.000), 0.958 (95%CI, 0.887–0.990), and 0.851 (95%CI, 0.753–0.921), respectively. Cut-off points for value differences in detecting subjectively improved gait were identical to mean −1.7 standard deviation (SD) for stride time, mean +1.6 SD for cadence, and mean +0.4 SD for ground floor reaction of control values from normal subjects. Objective gait test scores obtained using the tri-axial accelerometer can detect subjective gait improvements after CEA. When determining significant postoperative improvements in gait using a tri-axial accelerometer, optimal cut-off points for each test value can be defined.

## 1. Introduction

Severe stenosis of the cervical internal carotid artery (ICA) due to atherosclerosis causes cerebral ischemic stroke [1]. Surgical repair of this lesion including carotid endarterectomy (CEA) can prevent such stroke [1]. Furthermore, neurosurgeons, the families of patients and/or the patients themselves often subjectively report post-CEA improvements in neurological or neuropsychological symptoms and the reversibility of such symptoms, even among patients with a level of activities of daily living not requiring care assistance [2,3]. CEA occasionally restores cerebral perfusion that is reduced preoperatively due to ICA stenosis, and this recovery may result in improved brain functions [4,5]. Whereas numerous studies have investigated objective changes in cognitive function following CEA using neuropsychological testing in patients with activities of daily living not requiring care assistance [5,6,7,8], few have scientifically documented objective neurological changes after CEA in such patients [3]. Motor function is a basic and important neurological function in the human brain, and deficits in motor function are the most frequent symptoms following cerebral ischemic events due to cervical carotid artery stenosis. Human inspectors can assess severe or moderate motor deficits and the recovery from such deficits, but not recovery from slight motor deficits. CEA is usually applied for patients with activities of daily living not requiring any care assistance. Tests allowing objective detection of post-CEA recovery from slight motor deficits are therefore needed.

A previous study assessed hand motor function using the finger-tapping test and manipulation test before and after CEA and concluded that CEA does not have beneficial effects on motor function [9]. However, the sample size in that study was small, with only 20 patients. In addition, motor functions of the upper extremities are highly variable and complicated. Comprehensive, quantitative assessment of these functions may thus be difficult. Gait uses a repetitious sequence of lower limb motions to simultaneously move the body forward while maintaining stance stability [10]. A study using near-infrared spectroscopic topography demonstrated that walking bilaterally activates the medial primary sensorimotor cortices and supplementary motor areas [11]. Restoration of cerebral perfusion by CEA may lead to remyelination of the white matter in the contralateral frontal lobe as well as the ipsilateral cerebral hemisphere, resulting in postoperative improvements to brain function [12]. CEA can therefore improve neural functions in the regions that are activated while walking, leading to an improvement in gait function.

One study objectively and quantitatively assessed changes in gait performance before and after CEA using a motion analysis system consisting of infrared ray cameras, force plates and video cameras [3]. That study showed significant postoperative improvements in at least one gait test parameter in 88% of patients undergoing CEA [3]. However, evidence-based criteria for defining significant postoperative improvements in these quantitative gait tests were lacking, so differences between pre- and postoperative test scores for identifying clinically meaningful objective improvements in gait performance remained undetermined [3]. To investigate differences in postoperative changes to cerebral blood flow, cerebral metabolism, or cerebral neurotransmitter function among patients with and without significantly improved gait after surgery, evidence-based criteria for defining significant objective improvements in gait performance are necessary.

Portable devices for gait assessment such as tri-axial accelerometers have been developed and allow objective, quantitative assessment of gait function. These devices have now been applied in clinical studies [13,14,15,16,17,18]. In addition, the reliability of gait-related parameters obtained from tri-axial accelerometers has been confirmed with test-retest measurements [19].

The purpose of the present study was thus to determine whether objective gait test values obtained using a tri-axial accelerometer can detect subjective improvements in gait as determined by the patient after CEA in patients with activities of daily living not requiring care assistance. Results from this study suggest guidelines for determining significant objective improvements in gait function in such patients.

## 2. Materials and Methods

### 2.1. Patients

For this prospective observational study, patients were enrolled if they met these inclusion criteria: the affected cervical ICA showing ≥70% stenosis as calculated on magnetic resonance angiography, computed tomography, or arterial catheterization as described in the North American Symptomatic CEA Trial criteria [1]; no symptoms of ipsilateral carotid territory ischemia for ≥6 months prior to arrival at our hospital (asymptomatic), or symptoms of ipsilateral carotid territory ischemia 2 weeks to 6 months prior to arrival at our hospital (symptomatic); activities of daily living not requiring care assistance (modified Rankin disability scale score of 0–2) on arrival at the hospital; no sensory aphasia; scores for verbal and performance intelligence quotients of the Wechsler Adult Intelligence Scale-Revised of ≥80 [20]; and ability to walk smoothly for ≥100 m without any help or use of assistive devices on arrival at our hospital. Only patients who underwent CEA were eligible for this study.

The institutional ethics committee at our hospital assessed and approved this study protocol (H29-3; 21 April 2017). Written informed consent was obtained from each patient or a family member before enrollment in the study.

### 2.2. Assessment of Gait Function

Patients visited a neurologist, who was blinded to data from gait testing, ≤7 days before surgery and at 6 months after surgery. At the postoperative visit, the neurologist asked the patient whether gait had subjectively improved after the surgery when compared with the preoperative condition. Only when the patient clearly and promptly answered that gait had improved did the neurologist determine subjective gait improvement as present.

After examination by the neurologist before and at 6 months after surgery, an operator (S.F.), who was blinded to postoperative subjective assessments of gait, objectively and quantitatively assessed gait function using a tri-axial accelerometer (MG-M1110-HW; LSI Medience, Tokyo, Japan) in the same manner as described previously [19]. This accelerometer can measure tri-axial (vertical, anteroposterior, and mediolateral) acceleration by detecting limb and trunk movements at a sampling rate of 100 Hz during step-in and kick-off motions [19]. For each patient, the accelerometer was fixed at the L3 level of the spine with a nylon belt. Gait testing was performed on a straight 30 m walkway after instructing each patient to walk at their usual pace. The patient walked 16 m independently without any aides or assistive devices, and gait functions were measured over a 10 m segment between 3 m after the start of walking and 3 m before the end of walking. To mark the 10 m segment of the dataset, the operator (S.F.) manually pushed a button connected to the accelerometer with a cable at 3 m after the start of walking and 3 m before the end of walking while following the subject. This 16 m gait testing was repeated six times on each occasion.

From a dataset of each gait test in the 10 m segment, the following three parameters were calculated using commercial software (MG-M1110-HW; LSI Medience, Tokyo, Japan): stride time (time from initial contact of one foot to subsequent contact of the same foot, s); cadence (number of steps per 1 min, steps/min); and ground floor reaction (force from ground to the lower extremities during a step, ×9.8 m/s^2^) [19]. The six data points obtained from the six repeated measurements in each gait test parameter were averaged, and the average value was defined as the representative value for this gait test parameter [19]. Finally, the difference between representative values (postoperative value minus preoperative value) was calculated and defined as the Δ-value for each gait test parameter.

Prior to the start of the present study, control data for these gait tests were obtained from 36 healthy individuals tested on two independent occasions [19]. All methods for and analyses of data from gait testing were identical among controls and patients. Mean (± standard deviation (SD)) differences in each gait parameter between the two tests (second test value minus first test value), defined as the Δ-value in controls, were 0.006 ± 0.044 s for stride time, −0.4 ± 5.5 steps/min for cadence and −0.003 ± 0.062 × 9.8 m/s^2^ for ground floor reaction [19].

### 2.3. Surgery

All patients received antiplatelet drugs until the morning of the day of CEA, which was performed under general anesthesia. If hemispheric ischemia was indicated by intraoperative monitoring of transcranial cerebral oxygen saturation using near-infrared spectroscopy, an intraluminal shunt was implanted [21]. A 5000-IU heparin bolus was intravenously administered to the patient prior to ICA clamping.

### 2.4. Statistical Analysis

Univariate analysis with the Mann–Whitney *U* test or χ^2^ test was used to evaluate the relationship between a given parameter and subjective improvements in gait after surgery. The sequential backward elimination approach was used for logistic regression analysis of variables related to subjective improvements in gait. Exclusion of factors was halted when the *p*-value of the remaining variables reached <0.2. Changes between pre- and postoperative gait parameters were evaluated using the Wilcoxon signed-rank test. Differences in pre- and postoperative values or Δ-value for each gait test parameter between groups were examined using the Mann–Whitney *U* test. The accuracy of Δ-value for each gait test parameter in detecting subjective improvement in gait after surgery was assessed using receiver operating characteristic (ROC) curves. Area under the ROC curve (AUC) was used to determine the ability to discriminate between the presence and absence of subjectively improved gait. We used the method of Pepe and Longton [22] for pairwise comparisons of AUCs for gait test parameters. Binomial distributions were used to calculate the exact 95% confidence intervals (CIs) for sensitivity, specificity, and positive- and negative-predictive values, and differences in these values among gait test parameters were assessed with 95%CIs. All data are expressed as mean ± SD. All statistical analyses were performed on MedCalc version 17.9.7 (MedCalc Software bvba, Ostend, Belgium) with a significance level of *p* < 0.05.

## 3. Results

Over the course of 31 months, 85 patients satisfied the inclusion criteria. We excluded six patients: four patients declined to participate in the study; and two patients were not assessed after surgery. We thus analyzed 79 patients (76 men, 3 women) who underwent CEA and completed both pre- and postoperative gait testing.

Mean age of the 79 patients was 71 ± 7 years (range, 50–86 years). Comorbidities included hypertension (n = 69), diabetes mellitus (n = 24), and dyslipidemia (n = 60). Ischemic episodes in the ipsilateral carotid territory were present in 60 patients: 10 patients experienced transient ischemic attacks alone; and 50 patients experienced minor strokes with or without transient ischemic attacks. The time between the last attack and surgery ranged from 3 weeks to 23 weeks in these 60 symptomatic patients. Asymptomatic carotid stenosis was present in 19 patients. Contralateral ICA or middle cerebral artery occlusion or stenosis ≥50% was present in 30 patients. Mean degree of ICA stenosis was 87% (range, 70–99%), and mean duration of ICA clamping was 35 min (range, 16–47 min). Three patients required an intraluminal shunt.

Twelve patients (15%) reported subjectively improved gait after surgery, and 67 patients (85%) reported unimproved gait. Table 1 shows comparisons of each patient characteristic among these two groups using univariate analyses. The frequency of bilateral lesions (presence of contralateral ICA or middle cerebral artery occlusion or stenosis in addition to ICA stenosis ipsilateral to surgery) was significantly greater in patients with subjectively improved gait than in those without. No other parameters were associated with subjectively improved gait. The sequential backward elimination approach for logistic regression analysis showed that bilateral lesions were significantly associated with subjectively improved gait (95%CI, 1.04–14.58; *p* = 0.0430).

Table 2 shows pre- and postoperative values for each gait test parameter in all patients and in those with subjectively improved or unimproved gait after surgery. For all patients, while stride time (*p* = 0.1138) and cadence (*p* = 0.1568) did not differ pre- or postoperatively, ground floor reaction was significantly increased after surgery compared to before surgery (*p* = 0.0122). Stride time (*p* = 0.6620), cadence (*p* = 0.5615) and ground floor reaction (*p* = 0.5752) before surgery did not differ between patients with and without subjectively improved gait. While stride time (*p* = 0.0022) was significantly decreased and cadence (*p* = 0.0021) and ground floor reaction (*p* = 0.0022) were significantly increased after surgery compared to before surgery in patients with subjectively improved gait, values for all three gait test parameters did not differ pre- and postoperatively in patients without subjectively improved gait (*p* = 0.6250 for stride time; *p* = 0.6811 for cadence; *p* = 0.3252 for ground floor reaction). While stride time (*p* = 0.0062) and cadence (*p* = 0.0273) after surgery were significantly lower and higher, respectively, in patients with subjectively improved gait compared to those without subjectively improved gait, ground floor reaction (*p* = 0.1100) after surgery did not differ between these two patient groups.

Comparisons of Δ-values in each gait test parameter between patients with and without subjectively improved gait are shown in Figure 1. Patients with subjectively improved gait (−0.127 ± 0.056 s) showed a significantly lower Δ-value in stride time compared to patients without subjectively improved gait (0.002 ± 0.057 s; *p* < 0.0001). Patients with subjectively improved gait (cadence, 10.8 ± 5.8 steps/min; ground floor reaction, 0.078 ± 0.056 × 9.8 m/s^2^) showed significantly greater Δ-values for cadence and ground floor reaction than patients without subjectively improved gait (cadence, −0.321 ± 6.2 steps/min; ground floor reaction, 0.006 ± 0.056 × 9.8 m/s^2^) (cadence, *p* < 0.0001; ground floor reaction, *p* = 0.0001).

Based on Figure 1, the ROC curve for Δ-values in each gait test parameter for detecting subjectively improved gait were calculated as shown in Figure 2. AUCs for Δ-values in stride time, cadence, and ground floor reaction were 0.995 (95%CI, 0.945–1.000), 0.958 (95%CI, 0.887–0.990), and 0.851 (95%CI, 0.753–0.921), respectively. AUCs for Δstride time (difference between AUCs, 0.144; *p* = 0.0019) and Δcadence (difference between AUCs, 0.107; *p* = 0.0100) were significantly greater than that for Δground floor reaction. AUC did not differ between Δstride time and Δcadence (difference between AUCs, 0.037; *p* = 0.3147).

Table 3 shows the sensitivity, specificity, and positive- and negative-predictive values for Δ-values in each gait test parameter with the cut-off point lying closest to the upper left corner of the ROC curve for detecting subjectively improved gait. The cut-off point for Δ-values in detecting subjectively improved gait were identical to the mean −1.7 SDs (−0.081) for stride time, mean +1.6 SDs (9.0) for cadence, and mean +0.4 SDs (0.028) for ground floor reaction of the control values obtained from normal subjects [19]. The specificity and positive-predictive value were significantly greater for Δstride time or Δcadence than for Δground floor reaction. No values differed significantly between Δstride time and Δcadence.

## 4. Discussion

The present results suggest that objective gait test scores obtained using a tri-axial accelerometer can successfully detect subjective improvement in gait after CEA. When determining significant postoperative improvement in gait using a tri-axial accelerometer, optimal cut-off points for each test score can be defined.

In the present study, 15% of patients reported a subjectively improved gait after surgery. A previous study showed that approximately 10% of patients undergoing CEA exhibited significant increases in cerebral perfusion and neural receptor-binding potential compared to the preoperatively reduced levels resulting from ICA stenosis [4]. The prevalence of improvement in these two studies was comparable. Presence of contralateral ICA or middle cerebral artery occlusion or stenosis in addition to ICA stenosis ipsilateral to surgery was an independent predictor of subjectively improved gait. Cerebral perfusion in a patient with such conditions is often reduced [23]. Degree of postoperative increase in perfusion in the cerebral hemisphere ipsilateral to revascularization surgery is greater in patients with low cerebral perfusion before surgery than in those without [5,24]. This greater increase in cerebral perfusion may result in improvements to motor functions, including gait.

We selected stride time, cadence, and ground floor reaction among gait test parameters obtained using a tri-axial accelerometer. These representative parameters measure the basic features of walking in an individual. Stride time is also called “gait cycle”, and a reduction in this parameter implies smoother steps during walking [25]. An increase in cadence shows improvement in the motor balance of bilateral lower extremities [24]. An increase in ground force reaction implies increased muscular strength for lifting the legs [26]. Furthermore, these three gait test parameters reportedly show better correlations with higher intraclass correlation coefficient between test-retest measurements than other gait test parameters, such as number of steps, step length, velocity, coefficient of variance and assessment time in controls [19], indicating a degree of robustness for the three parameters used in the present study. Considering these findings, our group-rate analyses for the 79 patients (mean age, 71 years) with the ability to walk smoothly for ≥100 m without help or use of assistive devices showed that CEA usually improves ground floor reaction among gait functions. A previous study used motion analysis systems comprising of infrared cameras, force plates and video cameras, and demonstrated that cadence, speed, stride length, step time, and knee range of motion significantly improved after CEA [3]. Those findings partially differed from our results. All patients included in the previous study showed signs of gait disturbance before receiving CEA and mean age was 67 years [3]. These discrepancies in preoperative gait condition and age between the two studies may have contributed to the differences in results.

A previous study using a motion analysis system also showed significant postoperative improvement in at least one gait test parameter in 88% of patients undergoing CEA [3]. However, while quantitative gait tests using any device are regarded as objective measures of gait function, criteria for defining significant postoperative improvements in quantitative gait tests including motion analysis system remain unclear. Because patients undergoing CEA sometimes subjectively report postoperative improvements in gait, we defined subjective improvements based on feelings of significant postoperative improvements in gait. As a result, the degree of postoperative change in a score in each gait test parameter obtained using a tri-axial accelerometer differentiated patients with subjectively improved and unimproved gait after surgery. Cut-off points for the degree of postoperative change in each gait test parameter to detect subjective improvement in gait after surgery were also defined based on control values obtained from normal subjects.

ROC curve analyses showed degrees of postoperative change in stride time or cadence more accurately differentiated patients with subjectively improved and unimproved gait than that of ground floor reaction. These findings suggest that although CEA usually improves ground floor reaction, stride time and cadence are more suitable for detecting gait function significantly improved by CEA. In particular, sensitivity, specificity, and positive- and negative-predictive values for Δ-values in stride time for detecting subjectively improved gait were >85%. Stride time, therefore, seems to most strongly reflect the subjective assessment of postoperative improvement in gait and most accurately detects subjective improvement in gait after surgery when the optimal cut-off point for the test parameter is defined. In contrast, positive-predictive values for Δ-values in ground floor reaction for detecting subjectively improved gait were <40%, suggesting that this gait test parameter may be unsuitable for detecting significantly improved gait.

The present study possessed serious limitations that require consideration. Although we included patients without apparent motor deficits or impaired cognition, we could not estimate the extent of observer bias associated with subjective assessments of patient condition. Improvement or lack of improvement of gait as reported by patients themselves depends on the feelings of the individual, and this feeling is not always uniform among patients. Furthermore, subjective assessment was performed only once for each patient. The reliability and reproducibility of subjective assessments thus remains unknown. However, high AUCs for Δ-values in each gait test parameter for detecting subjectively improved gait (>0.85) suggest that observer bias might have only minimally affected postoperative subjective assessments of gait. To mark the 10 m segment of the dataset, an operator manually pushed a button connected to the accelerometer with a cable at 3 m after the start of walking and 3 m before the end of walking while following the subject. This manual determination of the 10 m segment represents another limitation. Although gait velocity is considered as a common test parameter, the present study did not analyze this gait parameter because of its low intraclass correlation coefficient between test-retest measurements in controls [19]. This is also a limitation. Novel wearable sensor systems with higher intraclass correlation coefficient between test-retest measurements have developed [27] and further studies using these systems would be of benefit. Lastly, stride time is calculated by dividing cadence by 60 and these two parameters are closely related to each other. This relationship means that the present study analyzed two independent gait parameters rather than three gait parameters.

## 5. Conclusions

Objective gait test scores obtained using a tri-axial accelerometer can detect subjective improvements in gait after CEA. When determining significant postoperative improvement in gait using such an accelerometer, optimal cut-off points for each test value can be defined. Further investigations regarding differences in postoperative changes of cerebral blood flow, cerebral metabolism, or cerebral neurotransmitter function among patients with and without tri-axial accelerometer-defined improved gait after surgery would be of benefit.

## Figures and Tables

**Figure 1 sensors-20-06590-f001:**
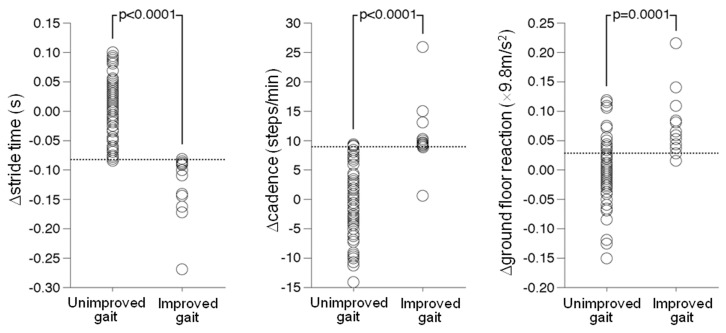
Comparisons of Δ-values (postoperative value minus preoperative value) in stride time (**left**), cadence (**middle**) and ground floor reaction (**right**) between patients with subjectively improved and unimproved gait.

**Figure 2 sensors-20-06590-f002:**
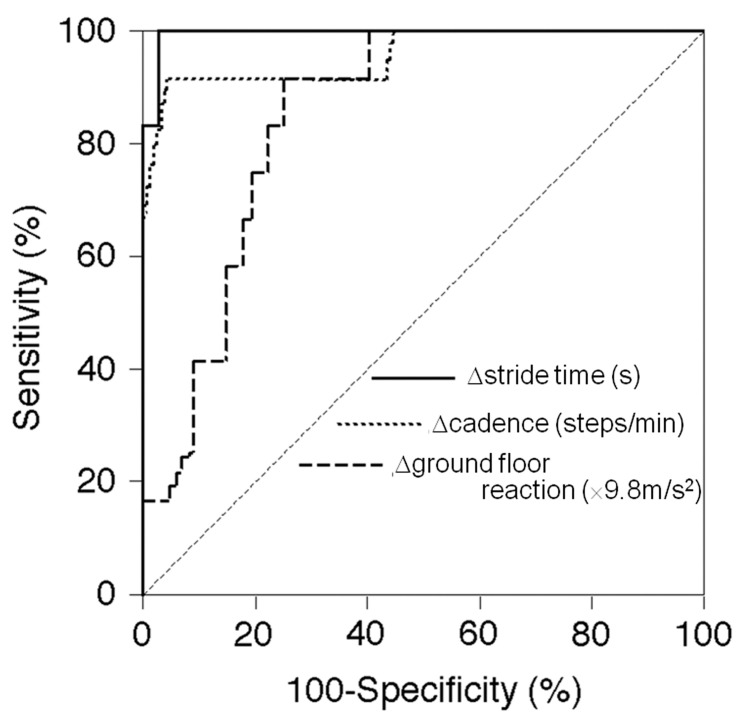
Receiver operating characteristic curve for Δ-values in each gait test parameter for detecting subjectively improved gait. Areas under the receiver operating characteristic curve for Δstride time (0.995) or Δcadence (0.958) were significantly greater than that for Δground floor reaction (0.851).

**Table 1 sensors-20-06590-t001:** Comparison of each patient characteristic between patients with subjectively improved and unimproved gait after surgery.

Variables	Subjectively Improved Gait (n = 12)	Subjectively Unimproved Gait (n = 67)	*p* Value
Age (years)	70.9 ± 8.0	70.9 ± 6.3	0.5288 †
Male sex	12 (100%)	64 (96%)	>0.9999 ‡
Hypertension	11 (91%)	58 (87%)	>0.9999 ‡
Diabetes mellitus	2 (17%)	22 (33%)	0.3276 ‡
Dyslipidemia	10 (83%)	50 (75%)	0.7203 ‡
Symptomatic lesion	7 (58%)	53 (79%)	0.1477 ‡
TIAs alone	2 (17%)	8 (12%)	0.6439 ‡
Minor strokes with/without TIAs	7 (58%)	43 (64%)	0.7510 ‡
Time between last attack and surgery [weeks]	14.3 ± 6.6	14.4 ± 4.9	0.9029 †
Bilateral lesions	8 (67%)	22 (33%)	0.0490 ‡
Degree of ICA stenosis [%]	88.8 ± 9.3	86.8 ± 9.4	0.5336 †
Duration of ICA clamping [min]	37.3 ± 4.5	34.7 ± 6.3	0.1468 †
Use of intraluminal shunt	0 (0%)	3 (4%)	>0.9999 ‡

TIA: transient ischemic attack; ICA: internal carotid artery. † Examined using the Mann-Whitney *U* test. ‡ Examined using the χ^2^ test.

**Table 2 sensors-20-06590-t002:** Pre- and postoperative values for each gait test parameter in all patients and in those with subjectively improved or unimproved gait after surgery.

	Stride Time (s)		Cadence (Steps/min)		Ground Floor Reaction (×9.8 m/s^2^)	
	Before Surgery	After Surgery	Before Surgery	After Surgery	Before Surgery	After Surgery
All patients	1.096 ± 0.120	1.078 ± 0.127	111.3 ± 11.7	112.7 ± 12.3	0.259 ± 0.071	0.276 ± 0.087
Patients with subjectively improved gait	1.100 ± 0.125	0.972 ± 0.171	110.7 ± 13.5	122.1± 19.0	0.255 ± 0.079	0.337 ± 0.138
Patients with subjectively unimproved gait	1.095 ± 0.120	1.097 ± 0.108	111.4 ± 11.5	111.0 ± 10.0	0.260 ± 0.070	0.265 ± 0.070

**Table 3 sensors-20-06590-t003:** Sensitivity, specificity, and positive- and negative-predictive values for Δ-value in each gait test parameter for detecting subjectively improved gait.

	*A* Stride Time	*B* Cadence	*C* Ground Floor Reaction	Statistical Significance by Comparison of 95% CIs		
				*A* versus *B*	*B* versus *C*	*C* versus *A*
Sensitivity (95%CI)	100% (100–100%)	92% (76–100%)	92% (76–100%)	No	No	No
Specificity (95%CI)	97% (93–100%)	96% (91–100%)	75% (64–85%)	No	Yes	Yes
Positive-predictive value (95%CI)	86% (67–100%)	79% (57–100%)	39% (21–57%)	No	Yes	Yes
Negative-predictive value (95%CI)	100% (100–100%)	98% (95–100%)	98% (95–100%)	No	No	No
Cut-off point	−0.081	9.0	0.028			

CI: confidence interval.

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
