# Peer review of "Comparison of Subjective and Objective Assessments on Improvement in Gait Function after Carotid Endarterectomy"

_sensors, 2020, doi:10.3390/s20226590_

Round 1

Reviewer 1 Report

Line 76, please specify date and number of the approval by Ethics Committee. 

References 2, 4, 5, 17, 19, 22, 23 are self-citations: is everyone necessary?

Line 243, reliability and reproducibility of these subjective assessments remain unknown: please discuss this item . 

Author Response

                   Responses to Reviewer 1

General comments: Minor spell check required

Response: The final revised version of our manuscript has been checked by a native English speaker.

Comment 1: Line 76, please specify date and number of the approval by Ethics Committee. 

Response and revisions:

Page 2, lines 44-45: We have added the requested information to the revised version of the text, as follows: “The institutional ethics committee at our hospital assessed and approved this study protocol (H29-3; April 21, 2017).”

Comment 2: References 2, 4, 5, 17, 19, 22, 23 are self-citations: is everyone necessary?

Response: References 4 and 23 can be deleted, but we consider References 2, 5, 17, 19, 22 as necessary.

Revisions:

We have deleted the following references: “4. Kojima, D.; Ogasawara, K.; Kobayashi, M.; Yoshida, K.; Kubo, Y.; Chida, K.; Oshida, S.; Yoshida, J.; Fujiwara, S;, Terasaki, K. Effects of uncomplicated carotid endarterectomy on cognitive function and brain perfusion in patients with unilateral asymptomatic severe stenosis of the internal carotid artery by comparison with unoperated patients. Neurol. Res. 2016, 38, 580-586.”and “23. Ando, T.; Shimada, Y.; Fujiwara, S.; Yoshida, K.; Kobayashi, M.; Kubo, Y.; Terasaki, K.; Ando, S.; Ogasawara, K. Revascularisation surgery improves cognition in adult patients with moyamoya disease. J. Neurol. Neurosurg. Psychiatry 2020, 91, 332-334.”

Comment 3: Line 243, reliability and reproducibility of these subjective assessments remain unknown: please discuss this item.

Response and revisions:

Page 8, lines 1-7: We have revised the text as follows: “Although we included patients without apparent motor deficits or impaired cognition, we could not estimate the extent of observer bias associated with subjective assessments of patient condition. Improvement or lack of improvement of gait as reported by patients themselves depends on the feelings of the individual and this feeling is not always uniform among patients. Further, subjective assessment was performed only once for each patient. The reliability and reproducibility of these subjective assessments thus remains unknown.”

Reference numbers: Based on the changes to the text, we have revised the reference numbers as follows: 54, 65, 76, 87, 98, 109, 2423, 2524.

Reviewer 2 Report

The authors used a tri-axial accelerometer to detect subjective gait improvements for individuals who received the carotid endarterectomy (CEA). The results suggested that the gait outcomes captured by the accelerometer (i.e. stride time, cadence, ground floor reaction) can differentiate those reported improved gait and unimproved gait after CEA. This manuscript provides interesting and novel information. Below are a few major comments:

  1. The scope and significance of the issue and/or the problem are not clear. For example, why did the authors choose individuals with the cervical internal carotid artery (ICA) as the study subjects? How popular/frequent do this population suffer from any gait problem? The authors only cited one 1999 reference relating to gait and those before/after CEA. Besides, the main justifications for doing this study are 1) the previous study used hand motor function, and 2) previous studies motion analysis, 3) both studies have little sample size.

  1. Gait function is not merely a motor function. The authors should review this field and consider integrating them in the background.

  1. Can the authors elaborate their statement “clear guidelines for determining significant objective improvements in gait performance are lacking, because such changes may, in part, reflect “practice effects” improvements in scores when patients undergo repeated testing”? The learning effect is rarely observed in walking, especially walking without conducting another cognitive task.

  1. Can the authors provide the raw gait information before and after the surgery, instead of only reporting the delta (changes) of the gait outcomes? We can then see the baseline of this cohort, both before and after the surgery.

  1. Only ~ 15% reported improved gait. I wonder whether gait is related to this surgery or may be related to other factors? Did the authors control them?

  1. Can the authors elaborate on the reasons they chose to report the three gait outcomes? Many other gait measures would also suggest a “smoother walking pattern”? The clinically meaningful gait measurements are typically gait speed, gait variability, and dual-task cost/gait performance, why didn’t the authors report any of them?

  1. What is the minimally important difference for the gait outcomes the authors reported? Are they similar to the cut-off values they found in this study? Would be nice to discuss this in the discussion section.

  1. Test-retest reliability for the motion analysis systems is typically high. And in fact, the gait measure from the motion analysis is sensitive and usually serves as a gold standard. Please explain the rationale of the statement in the content: “The robustness of gait test parameters in test-retest measurements using motion analysis systems remains unclear.”

  1. * and ** (Asterisk) signs are typically used for statistical significance. Would recommend the authors select different symbols for Table 1, to avoid confusion.

  1. Sounds like the gait was measured three time-points: pre-op (7 days before the surgery), immediately after the surgery, and 6 months after the surgery. How did the authors calculate the delta (post-pre)? which post?

  1. Looks like the authors measured the gait twice for a set of healthy adults (i.e. control data). If the authors chose to mention this, please provide the methods for this healthy control portion, and discuss the differences among healthy control, those reported improved gait, and those reported unimproved gait.

  1. How did the authors manage/analyze the gait data? Did the authors remove data from the first 3m and last 3m of the gait assessment?

  1. In the limitation, what observer bias did the authors refer to? Did the authors use any blinding protocol? How did the observer bias influence the study?

  1. What exactly is the subjective gait assessment? How did the author ask the patient? Can the authors report the script they asked the patients?

Author Response

                   Responses to Reviewer 2

General comments: Moderate English changes required

Response: The final revised version of our manuscript has been checked by a native English speaker.

Comment 1: The scope and significance of the issue and/or the problem are not clear. For example, why did the authors choose individuals with the cervical internal carotid artery (ICA) as the study subjects? How popular/frequent do this population suffer from any gait problem? The authors only cited one 1999 reference relating to gait and those before/after CEA. Besides, the main justifications for doing this study are 1) the previous study used hand motor function, and 2) previous studies motion analysis, 3) both studies have little sample size.

Response: Neurosurgeons, the families of patients and/or the patients themselves often subjectively report post-carotid endarterectomy (CEA) improvements in neurological or neuropsychological symptoms and the reversibility of such symptoms, even among patients with a level of activities of daily living not requiring any care assistance. While numerous studies have investigated objective changes in cognitive function following CEA using neuropsychological testing in patients with activities of daily living not requiring any care assistance, few have scientifically documented objectively neurological changes after CEA in such patients. Human inspectors can assess severe motor deficits and recovery of such deficits, but not recovery of fine motor deficits. CEA is usually applied for patients with activities of daily living not requiring any care assistance. Testing for objectively detecting post-CEA recovery of fine motor deficits is thus needed. In the finger-tapping test, the manipulation test, and the motion analysis system, clear guidelines for determining significant objective improvements in gait performance are lacking, and differences between pre- and postoperative test scores that can be regarded as representing significant objective improvements in gait performance remain undetermined.

The purpose of the present study was thus to determine whether objective gait test values obtained using a tri-axial accelerometer can detect subjective improvements in gait as determined by the patient after CEA among patients with activities of daily living not requiring care assistance. Results from this study suggest guidelines for determining significant objective improvements in gait function among such patients.

Revisions:

Pages 1-2, paragraph 1 in the “1. Introduction: We have revised the paragraph as follows: “Severe stenosis of the cervical internal carotid artery (ICA) due to atherosclerosis causes cerebral ischemic stroke [1]. Surgical repair of this lesion including carotid endarterectomy (CEA) can prevent such stroke [1]. Further, neurosurgeons, the families of patients and/or the patients themselves often subjectively report post-CEA improvements in neurological or neuropsychological symptoms and the reversibility of such symptoms, even among patients with a level of activities of daily living not requiring care assistance [2,3]. CEA occasionally restores cerebral perfusion that is reduced preoperatively due to ICA stenosis, and this recovery may result in improved brain functions [4,5]. Whereas numerous studies have investigated objective changes in cognitive function following CEA using neuropsychological testing in patients with activities of daily living not requiring care assistance [5-8], few have scientifically documented objective neurological changes after CEA in such patients [3]. Motor function is a basic and important neurological function in the human brain and deficits in motor function are the most frequent symptoms following cerebral ischemic events due to cervical carotid artery stenosis. Human inspectors can assess severe motor deficits and recovery of such deficits, but not recovery of fine motor deficits. CEA is usually applied for patients with activities of daily living not requiring any care assistance. Tests allowing objective detection of post-CEA recovery of fine motor deficits are thus needed.”

Page 2, lines 18-21: We have revised the sentence as follows: “Further, clear guidelines for determining significant objective improvements in gait performance were lacking in that study, so differences between pre- and postoperative test scores for identifying significant objective improvements in gait performance remained undetermined.”

Page 2, lines 26-30: We have revised the paragraph as follows: “The purpose of the present study was thus to determine whether objective gait test values obtained using a tri-axial accelerometer can detect subjective improvements in gait as determined by the patient after CEA in patients with activities of daily living not requiring care assistance. Results from this study suggest guidelines for determining significant objective improvements in gait function in such patients.”

Page 2, lines 33-42: We have revised the text as follows: “For this prospective observational study, patients were enrolled if they met these inclusion criteria: the affected cervical ICA showing ≥70% stenosis as calculated on magnetic resonance angiography, computed tomography, or arterial catheterization as described in the North American Symptomatic CEA Trial criteria [1]; no symptoms of ipsilateral carotid territory ischemia for ³6 months prior to arrival at our hospital (asymptomatic), or symptoms of ipsilateral carotid territory ischemia 2 weeks to 6 months prior to arrival at our hospital (symptomatic); activities of daily living not requiring care assistance (modified Rankin disability scale score of 0–2) on arrival at the hospital; no sensory aphasia; scores for verbal and performance intelligence quotients of the Wechsler Adult Intelligence Scale-Revised of ≥80 [18]; and ability to walk smoothly for ≥100 m without any help or use of assistive devices on arrival at our hospital.”

Comment 2: Gait function is not merely a motor function. The authors should review this field and consider integrating them in the background.

Response and revisions:

Page 2, lines 11-16: We have revised the text as follows: “Gait uses a repetitious sequence of lower limb motions to simultaneously move the body forward while maintaining stance stability [10]. Because each sequence involves a series of interactions between two multisegmented lower extremities and the total body mass, identification of the numerous events involved necessitates a view of gait from several different aspects [10]. One study assessed changes in gait performance before and after CEA using a motion analysis system consisting of infrared ray cameras, force plates and video cameras [3].”

Page 9, lines 1-2 We have added the following reference: “10. Perry, J.; Burnfield, J.M. Gait Analysis: Normal and Pathological Function. New Jersey, Slack Incorporated, 2010.”

Comment 3: Can the authors elaborate their statement “clear guidelines for determining significant objective improvements in gait performance are lacking, because such changes may, in part, reflect “practice effects” improvements in scores when patients undergo repeated testing”? The learning effect is rarely observed in walking, especially walking without conducting another cognitive task.

Response: As suggested, “practice effects” were also not observed in gait test values obtained with the tri-axial accelerometer used in the present study [1]. We have deleted the phrase regarding “practice effects”.

  1. Fujiwara, S.; Sato, S.; Sugawara, A.; Nishikawa, Y.; Koji, T.; Nishimura, Y.; Ogasawara, K. The coefficient of variation of step time can overestimate gait abnormality: test-retest reliability of gait-related parameters obtained with a tri-axial accelerometer in healthy subjects. Sensors (Basel) 2020, 20, 577.

Revisions:

Page 2, lines 18-21: We have revised the sentence as follows: “Further, clear guidelines for determining significant objective improvements in gait performance were lacking in that study, so differences between pre- and postoperative test scores for identifying significant objective improvements in gait performance remained undetermined.”

Comment 4: Can the authors provide the raw gait information before and after the surgery, instead of only reporting the delta (changes) of the gait outcomes? We can then see the baseline of this cohort, both before and after the surgery.

Response: We have already provided the raw gait information before and after the surgery in paragraph 4 of the original manuscript, as follows: “For all patients, while stride time (P = 0.1138) and cadence (P = 0.1568) did not differ between preoperatively (stride time, 1.096 ± 0.120 s; cadence, 111.3 ± 11.7 steps/min) and postoperatively (stride time, 1.078 ± 0.127 s; cadence, 112.7 ± 12.3 steps/min), ground floor reaction was significantly increased after surgery (0.276 ± 0.087 ×9.8 m/s2) compared to before surgery (0.259 ± 0.071 ×9.8 m/s2; P = 0.0122).”

In the revised manuscript, we have added statistical comparisons of preoperative value for each gait test parameter between patients with and without subjectively improved gait. As a result, no gait test parameters before surgery showed significant differences between patient subgroups.

Revisions:

Page 3, lines 44-45: We have revised the sentence as follows: “Differences in preoperative value or D-value for each gait test parameter between groups were examined using the Mann-Whitney U test.”

Page 5, lines 3-8: We have added the following sentence: “Stride time (1.100 ± 0.125 s for subjectively improved gait; 1.095 ± 0.120 s for subjectively not improved gait; P = 0.6620), cadence (110.7 ± 13.5 steps/min for subjectively improved gait; 111.4 ± 11.5 steps/min for subjectively not improved gait; P = 0.5615) and ground floor reaction (0.255 ± 0.079 ×9.8 m/s2 for subjectively improved gait; 0.260 ± 0.070 ×9.8 m/s2 for subjectively not improved gait; P = 0.5752) before surgery did not differ between patients with and without subjectively improved gait.”

Comment 5: Only ~ 15% reported improved gait. I wonder whether gait is related to this surgery or may be related to other factors? Did the authors control them?

Response: As suggested, only 15% of patients reported subjectively improved gait. In the Introduction section, we described that CEA occasionally restores cerebral perfusion that is reduced preoperatively due to ICA stenosis, and this recovery may result in improved brain functions [5]. The study in Reference 5 measured cerebral perfusion and neural receptor binding potential using single-photon emission computed tomography. Approximately 10% of patients undergoing CEA were demonstrated to exhibit significant increases in cerebral perfusion and neural receptor binding potential, which were reduced preoperatively due to ICA stenosis. In the present study, although we did not measure these two values, the prevalence of these two studies (15% of patients who reported subjectively improved gait versus 10% of patients who postoperatively exhibited significant increases in cerebral perfusion and neural receptor binding potential) was comparable. As described in the Discussion section (paragraph 2), presence of contralateral ICA or middle cerebral artery occlusion or stenosis in addition to ICA stenosis ipsilateral to surgery was an independent predictor of subjectively improved gait. Cerebral perfusion in a patient with such conditions is often reduced. Degree of postoperative increase in perfusion in the cerebral hemisphere ipsilateral to revascularization surgery is greater in patients with low cerebral perfusion before surgery than in those without. This greater increase in cerebral perfusion may result in improvements to motor functions, including gait.

  1. Chida, K.; Ogasawara, K.; Aso, K.; Suga, Y;, Kobayashi, M.; Yoshida, K.; Terasaki, K.; Tsushina, E.; Ogawa, A. Postcarotid endarterectomy improvement in cognition is associated with resolution of crossed cerebellar hypoperfusion and increase in 123I-iomazenil uptake in the cerebral cortex: a SPECT study. Cerebrovasc. Dis. 2010, 29, 343–351.

Revisions:

Page 7, lines 1-5: We have revised the text as follows: “In the present study, 15% of patients reported subjectively improved gait after surgery. A previous study showed that approximately 10% of patients undergoing CEA exhibit significant increases in cerebral perfusion and neural receptor-binding potential compared to the preoperatively reduced levels resulting from ICA stenosis [4]. The prevalence of improvement in these two studies was comparable.”

Comment 6: Can the authors elaborate on the reasons they chose to report the three gait outcomes? Many other gait measures would also suggest a “smoother walking pattern”? The clinically meaningful gait measurements are typically gait speed, gait variability, and dual-task cost/gait performance, why didn’t the authors report any of them?

Response: As suggested, we selected stride time, cadence, and ground floor reaction among gait test parameters obtained using a tri-axial accelerometer. These representative parameters measure the basic features of walking in an individual. Stride time is also called “gait cycle”, and a reduction in this parameter implies smoother steps during walking. An increase in cadence shows improved motor balance in bilateral lower extremities. An increase in ground force reaction implies increased muscular strength for lifting the legs. Further, these three gait test parameters reportedly show better correlations with higher intraclass correlation coefficient between test-retest measurements than other gait test parameters, such as number of steps, step length, velocity, coefficient of variance and assessment time in controls [1], indicating a robustness in these three parameters used in the present study.

  1. Fujiwara, S.; Sato, S.; Sugawara, A.; Nishikawa, Y.; Koji, T.; Nishimura, Y.; Ogasawara, K. The coefficient of variation of step time can overestimate gait abnormality: test-retest reliability of gait-related parameters obtained with a tri-axial accelerometer in healthy subjects. Sensors (Basel) 2020, 20, 577.

Revisions:

Page 7, lines 12-21: We have revised the text as follows: “We selected stride time, cadence, and ground floor reaction among gait test parameters obtained using a tri-axial accelerometer. These representative parameters measure the basic features of walking in an individual. Stride time is also called “gait cycle”, and a reduction in this parameter implies smoother steps during walking [23]. An increase in cadence shows improvement in the motor balance of bilateral lower extremities [24]. An increase in ground force reaction implies increased muscular strength for lifting the legs [24]. Further, these three gait test parameters reportedly show better correlations with a higher intraclass correlation coefficient between test-retest measurements than other gait test parameters, such as number of steps, step length, velocity, coefficient of variance and assessment time in controls [17], indicating a degree of robustness for the three parameters used in the present study.”

Comment 7: What is the minimally important difference for the gait outcomes the authors reported? Are they similar to the cut-off values they found in this study? Would be nice to discuss this in the discussion section.

Response and revisions:

Page 7, lines 42-52: We have revised the paragraph as follows: “ROC curve analyses showed degrees of postoperative change in stride time or cadence more accurately differentiated patients with subjectively improved and unimproved gait than that in ground floor reaction. These findings suggest that although CEA usually improves ground floor reaction, stride time and cadence are more suitable for detecting gait function significantly improved by CEA. In particular, sensitivity, specificity, and positive- and negative-predictive values for Δ-values in stride time for detecting subjectively improved gait were >85%. Stride time thus seems to most strongly reflect the subjective assessment of postoperative improvement in gait and most accurately detect subjective improvement in gait after surgery when the optimal cut-off point for the test parameter is defined. In contrast, positive-predictive values for D-values in ground floor reaction for detecting subjectively improved gait were <40%, suggesting that this gait test parameter may be unsuitable for detecting significantly improved gait.”

Comment 8: Test-retest reliability for the motion analysis systems is typically high. And in fact, the gait measure from the motion analysis is sensitive and usually serves as a gold standard. Please explain the rationale of the statement in the content: “The robustness of gait test parameters in test-retest measurements using motion analysis systems remains unclear.”

Response: We are sorry for our insufficient discussion in the original manuscript. We have confirmed high test-retest reliability for the motion analysis systems. Our group-rate analyses for the 79 patients (mean age, 71 years) with the ability to walk smoothly for ≥100 m without any help or use of assistive devices showed that CEA usually improves ground floor reaction among gait functions. A previous study using motion analysis systems comprising infrared cameras, force plates and video cameras demonstrated that cadence, speed, stride length, step time, and knee range of motion significantly improved after CEA [1]. Those findings partially differed from our results. All patients who were included in the previous study had signs of gait disturbance before receiving CEA and a mean age of 67 years [1]. These discrepancies of preoperative gait condition and age among the two studies may have contributed to the partially different results.

  1. Kim, G.E.; Yoo, J.Y.; Cho, Y.P.; Ha, S.B. Gait improvement after carotid endarterectomy. J. Stroke Cerebrovasc. Dis. 1999, 8, 307-311.

Revisions:

Page 7, lines 21-30: We have revised the text as follows: “Considering these findings, our group-rate analyses for the 79 patients (mean age, 71 years) with the ability to walk smoothly for ≥100 m without help or use of assistive devices showed that CEA usually improves ground floor reaction among gait functions. A previous study used motion analysis systems comprising infrared cameras, force plates and video cameras, and demonstrated that cadence, speed, stride length, step time, and knee range of motion significantly improved after CEA [3]. Those findings partially differed from our results. All patients included in the previous study showed signs of gait disturbance before receiving CEA and mean age was 67 years [3]. These discrepancies in preoperative gait condition and age between the two studies may have contributed to the differences in results.”

Comment 9: * and ** (Asterisk) signs are typically used for statistical significance. Would recommend the authors select different symbols for Table 1, to avoid confusion.

Response and revisions: We have revised Table 1 as suggested.

Comment 10: Sounds like the gait was measured three time-points: pre-op (7 days before the surgery), immediately after the surgery, and 6 months after the surgery. How did the authors calculate the delta (post-pre)? which post?

Response: Patients visited a neurologist ≤7 days before surgery and at 6 months after surgery. At the postoperative visit, the neurologist asked the patient whether gait had subjectively improved since recovery from surgery when compared with the preoperative condition. After examination by the neurologist before and at 6 months after surgery, gait function was objectively, quantitatively assessed using a tri-axial accelerometer. Patients did not undergo measurement of gait function using a tri-axial accelerometer immediately after surgery.

Revisions:

Page 3, lines 4-7: We have revised the sentence as follows: “After examination by the neurologist before and at 6 months after surgery, an operator (S.F.), who was blinded to postoperative subjective assessments of gait, objectively and quantitatively assessed gait function using a tri-axial accelerometer (MG-M1110-HW; LSI Medience, Tokyo, Japan) in the same manner as described previously [17].”

Comment 11: Looks like the authors measured the gait twice for a set of healthy adults (i.e. control data). If the authors chose to mention this, please provide the methods for this healthy control portion, and discuss the differences among healthy control, those reported improved gait, and those reported unimproved gait.

Response: Prior to the start of the present study, control data for these gait tests were obtained from 36 healthy individuals tested on two independent occasions [1]. The methods and data analyses of this gait testing were identical among controls and patients, and also among patients with and without subjectively improved gait after surgery. We have therefore not discussed the differences in methods among healthy controls, those who reported improvement of gait, and those who reported no improvement of gait.

  1. Fujiwara, S.; Sato, S.; Sugawara, A.; Nishikawa, Y.; Koji, T.; Nishimura, Y.; Ogasawara, K. The coefficient of variation of step time can overestimate gait abnormality: test-retest reliability of gait-related parameters obtained with a tri-axial accelerometer in healthy subjects. Sensors (Basel) 2020, 20, 577.

Revisions:

Page 3, lines 26-31: We have revised the text as follows: “Prior to the start of the present study, control data for these gait tests were obtained from 36 healthy individuals tested on two independent occasions [17]. All methods for and analyses of data from gait testing were identical among controls and patients. Mean (± standard deviation (SD)) differences in each gait parameter between the two tests (second test value - first test value), defined as the D-value in controls, were 0.006 ± 0.044 s (mean ± standard deviations (SDs)) for stride time, -0.4 ± 5.5 steps/min for cadence and -0.003 ± 0.062 ×9.8 m/s2 for ground floor reaction [17].”

Comment 12: How did the authors manage/analyze the gait data? Did the authors remove data from the first 3m and last 3m of the gait assessment?

Response: The patient walked 16 m independently without any aides or assistive devices, and gait functions were measured over a 10-m segment between 3 m after the start of walking and 3 m before the end of walking. To mark the 10-m segment of the dataset, an operator (S.F.) manually pushed a button connected to the accelerometer with a cable at 3 m after the start of walking and 3 m before the end of walking while following the subject. This 16-m gait testing was repeated six times on each occasion. This manual determination of the 10-m segment also represents a limitation. We have mentioned this limitation in the Discussion section.

Revisions:

Page 3, lines 14-16: We have added the following sentence: “To mark the 10-m segment of the dataset, the operator (S.F.) manually pushed a button connected to the accelerometer with a cable at 3 m after the start of walking and 3 m before the end of walking while following the subject.”

Page 8, line 1: We have revised the text as follows: “The present study possessed serious limitations that require consideration.”

Page 8, lines 9-12: We have added the following sentences: “To mark the 10-m segment of the dataset, an operator manually pushed a button connected to the accelerometer with a cable at 3 m after the start of walking and 3 m before the end of walking while following the subject. This manual determination of the 10-m segment represents another limitation.”

Comment 13: In the limitation, what observer bias did the authors refer to? Did the authors use any blinding protocol? How did the observer bias influence the study?

Response: Although we included patients without apparent motor deficits and impaired cognition, we could not estimate the extent of observer bias associated with subjective assessments of patient condition regarding improvement or no improvement of gait as reported by the patients themselves depending on their feelings, and this feeling was not always consistent among patients. Further, subject assessment was performed only once for each patient. The reliability and reproducibility of these subjective assessments thus remain unknown. However, high AUCs for D-values in each gait test parameter for detecting subjectively improved gait (>0.85) suggested that observer bias might minimally affect postoperative subjective assessments of gait.

Revisions:

Page 8, lines 1-9: We have revised the text as follows: “The present study possessed serious limitations that require consideration. Although we included patients without apparent motor deficits or impaired cognition, we could not estimate the extent of observer bias associated with subjective assessments of patient condition. Improvement or lack of improvement of gait as reported by patients themselves depends on the feelings of the individual and this feeling is not always uniform among patients. Further, subjective assessment was performed only once for each patient. The reliability and reproducibility of these subjective assessments thus remains unknown. However, high AUCs for D-values in each gait test parameter for detecting subjectively improved gait (>0.85) suggest that observer bias might have only minimally affected postoperative subjective assessments of gait.”

Comment 14: What exactly is the subjective gait assessment? How did the author ask the patient? Can the authors report the script they asked the patients?

Response: Patients visited a neurologist (who was blinded to data from gait tests) ≤7 days before surgery and at 6 months after surgery. At the postoperative visit, the neurologist asked the patient whether gait had subjectively improved after the surgery when compared with the preoperative condition. Only when the patient clearly and promptly responded that their gait had improved did the neurologist determine that gait had subjectively improved. After examination by the neurologist before and at 6 months after surgery, an operator (S.F.) who was blinded to postoperative subjective assessments of gait objectively and quantitatively assessed gait function using the tri-axial accelerometer (MG-M1110-HW; LSI Medience, Tokyo, Japan) in the same manner as described previously [1]. The operator (S.F.) is not a medical doctor, but a neuroscientist (Ph.D.).

  1. Fujiwara, S.; Sato, S.; Sugawara, A.; Nishikawa, Y.; Koji, T.; Nishimura, Y.; Ogasawara, K. The coefficient of variation of step time can overestimate gait abnormality: test-retest reliability of gait-related parameters obtained with a tri-axial accelerometer in healthy subjects. Sensors (Basel) 2020, 20, 577.

Revisions:

Page 2, line 48-page 3, line 7: We have revised the text as follows: “Patients visited a neurologist, who was blinded to data from gait testing, ≤7 days before surgery and at 6 months after surgery. At the postoperative visit, the neurologist asked the patient whether gait had subjectively improved after the surgery when compared with the preoperative condition. Only when the patient clearly and promptly answered that gait had improved did the neurologist determine subjective gait improvement as present.

After examination by the neurologist before and at 6 months after surgery, an operator (S.F.), who was blinded to postoperative subjective assessments of gait, objectively and quantitatively assessed gait function using a tri-axial accelerometer (MG-M1110-HW; LSI Medience, Tokyo, Japan) in the same manner as described previously [17].”

Other revisions: Page 2, lines 15-16: We have revised the text as follows: “One study assessed changes in gait performance before and after CEA using a motion analysis system consisting of infrared ray cameras, force plates and video cameras [3].”

Reference numbers: We have revised the reference numbers as follows: 5→4, 6→5, 7→6, 8→7, 9→8, 10→9, 24→23, 25→24.

Round 2

Reviewer 2 Report

  1. The authors still did not answer how common CEA influences gait. If the purpose was to “determine whether objective gait test values obtained using a tri-axial accelerometer can detect subjective improvements in gait as determined by the patient after CEA”. The introduction should provide background related to the issues and the gap of this field. Instead, the author only provided one 1999 paper that studied gait analysis after CEA.

Fine motor function (e.g., eating, writing, using a tool) and care assistance are both outside of the scope of this study. Gait is a gross motor function, not fine motor function. Those who have gait problems may or may not need care assistance. Don’t understand why the authors mentioned fine motor and care assistance in the introduction. The significance and scientific merit of the article (e.g. how CEA influences gait, why do we need to conduct this study, why do we need to measure gait after CEA) can’t be found in the introduction section.

  1. The authors did not mention the link between gait and brain, which sounds like the main foundation of this study. The author wrote “Because each sequence involves a series of interactions between two multisegmented low extremities and body mass…”, suggesting the authors viewed gait as a biomechanical movement. I would recommend the authors do a quick literature review about recent gait studies, especially how gait links to brain functions.

  1. Please be specific throughout the manuscript. For example, what are “several different aspects”?

  1. The data suggested no significant differences in all the gait outcomes (i.e. stride time, cadence, ground reaction force) between subjectively improved gait and not-improved gait for either before or after surgery. Not sure whether the delta-value is then meaningful?

Author Response

This box accurately dose not reflect deletion lines and color of fonts. So, I have uploaded our replies to Reviewer 2.

This manuscript is a resubmission of an earlier submission. The following is a list of the peer review reports and author responses from that submission.